# Conditionally Integrable Nonlinear Diffusion with Diffusivity 1/$u$

**Philip Broadbridge [1,2,]* and Joanna M. Goard [2]**

1   Department of Mathematics and Statistics, La Trobe University, Bundoora, VIC 3086, Australia
2   School of Mathematics and Applied Statistics, University of Wollongong,
    Wollongong, NSW 2522, Australia; joanna@uow.edu.au
*   Correspondence: P.Broadbridge@latrobe.edu.au

**Abstract:** An explicit mapping is given from the space of general complex meromorphic functions to a space of special time-dependent solutions of the 1 + 2-dimensional nonlinear diffusion equation with diffusivity depending on concentration as $D = 1/u$. These solutions have constant-flux boundary conditions. Some simple examples are constructed, including that of a line source enclosed by a cylindrical barrier. This has direct application to electron diffusion in a laser-heated plasma.

**Keywords:** nonlinear diffusion; conditional integrability; electron diffusion; laser-heated plasma

## 1. Introduction

There are relatively few applicable nonlinear partial differential equations (PDEs) for which the full infinite-dimensional manifold of classical solutions can be constructed explicitly. These special integrable equations have an infinite-dimensional Lie symmetry algebra that includes a general solution of a linear PDE that relates to the original PDE either by a Darboux equivalence transformation or by an inverse scattering transform [1]. Some non-integrable PDEs have a finite-dimensional solvable Lie symmetry algebra, which leads to a finite-dimensional manifold of explicit solutions [2]. Many such symmetry reductions have been enumerated and classified [3,4]. Less familiar are the unusual PDEs that have an infinite-dimensional symmetry algebra, but are nonetheless non-integrable. These may lead to an infinite-dimensional manifold of explicit solutions that is a proper subset of the full solution manifold, not covering the most general allowable boundary data and/or initial values. The PDE that is studied here is known to have an infinite-dimensional symmetry group [5], but the corresponding class of exact solutions has not been given fully. The class of invariant solutions will be constructed here and applied to electron diffusion in a plasma. Several examples of initial-boundary problems will be solved in the classical sense of constructing a smooth solution. One of the simpler examples is that of zero initial density, prescribed constant flux condition at the inner cylindrical surface representing a laser beam and Neumann zero-flux condition at an outer concentric cylindrical surface representing a container that is in practice effected by a magnetic field. As has long been familiar from similarity solutions of diffusion equations (e.g., [6]), there may be point sources or sinks at a finite number of singularities, but there are domains with no singularities and with meaningful flux boundary conditions.

The focus here is on the nonlinear diffusion equation,

$$u_t = \nabla \cdot \left[ \frac{D_0}{u} \nabla u \right]. \tag{1}$$

A subscript $t, x, y, r$ or $\phi$ will denote a partial derivative with respect to a time coordinate, or Cartesian space coordinate, or polar coordinate, respectively.

This partial differential equation (PDE) with $\nabla$ representing the spatial gradient $\partial/\partial x^i$ applies to diffusion of electrons in a fully-ionised plasma. For n-dimensional flows of electrons, $u$ will be the electron density (dimensions $L^{-n}$). Near equilibrium, the electron diffusion coefficient depends on electron density $u$ and Boltzmann temperature $k_B T$ according to $D = D_0/u$, where $D_0 = k_B T K_0(u)$, with $K_0$ weakly increasing as $u$ varies by several orders of magnitude. For the theoretical formulation of $D_0$ and the measurement of the temperature-independent transport modification coefficient $K_0$, the reader is referred to [7]. For the purpose of estimating diffusive transport, the nonlinear diffusivity is approximated by $D = D_0/u$ with $D_0$ constant. Note that whereas $[D] = L^2 T^{-1}$ (the standard dimensions of diffusivity),

$$[D_0] = [D][u] = L^{2-n} T^{-1}. \tag{2}$$

For the remainder of this article, attention will be restricted to one time and two space dimensions. As the considered similarity solutions are classical, the PDE will be defined on functions that are differentiable in time and twice differentiable in space for $t \in \mathbb{R}^+$ and $\mathbf{r} \in \mathbb{R}^2$, with the possible exception of a finite number of singular points. In two space dimensions, Equation (1) has the special property of having an infinite-dimensional Lie point symmetry group, even though it is not integrable. In fact, the symmetry group includes the infinite-dimensional group of conformal maps that can be parametrised by a free complex analytic function. Another example of a non-integrable PDE with an infinite-dimensional classical symmetry group was given in Section 10.9 of [4]. An example of a non-integrable parabolic equation with a nonlinear reaction term and infinite-dimensional non-classical symmetry group was given in [8]. From a pair of conjugate harmonic functions, one may in principle, but not always in practice, construct a full set of invariants that reduces Equation (1) from three to two independent variables. There is no guarantee that the reduced PDE will be further reducible by a larger solvable symmetry algebra. A full symmetry classification of one-dimensional energy diffusion in a laser-heated plasma has been given in [9]. However, the one-dimensional model gives little indication of the especially large symmetry group resulting from a nonlinear diffusivity of the form $D = u^{-1}$ in two spatial dimensions. In fact, among all nonlinear diffusion equations in two space dimensions, the nonlinear diffusion Equation (5) has the largest symmetry group [5].

It would be of some advantage to have an explicit mapping from the class of conjugate solutions of Laplace's equation in two dimensions, to a special class of t-dependent solutions of the PDE (1) in three independent variables. A version of that mapping, directly related to Liouville's linearisation of exponential nonlinear steady reaction-diffusion, is given in Section 3. In Section 4, the boundary conditions for the class of solutions is clarified, and some special time-dependent solutions are derived from familiar classical solutions to Laplace's equation, including the point source, point vortex and image pair. The latter maintains for (1), a no-flow boundary condition across a specified plane. In Section 2, some preliminary ground work is done in rescaling Equation (1) in the context of electron diffusion, then recounting the Lie symmetry group.

## 2. Symmetry Group, after Rescaling

For two-dimensional flows embedded in three-dimensional Euclidean space, it is implicitly assumed that the number of particles has been integrated over a unit of depth in the remaining normal direction. In the application to diffusing electrons, $u$ is the particle number density per unit area ($[u] = L^{-2}$). From (2), $1/D_0$ has the dimensions of time, being the small time scale for diffusion over a typical separation distance between particles. This separation distance is $u_0^{-1/2}$ where $u_0$ is a representative areal particle density. In terms of dimensionless variables,

$$u^* = u/u_0, \quad (x^*, y^*) = u_0^{1/2}(x, y), \quad \nabla^* = u_0^{-1/2}\nabla, \quad t^* = D_0 t, \tag{3}$$

$$u_{t^*}^* = \nabla^* \cdot [\frac{1}{u^*}\nabla^* u^*]. \tag{4}$$

Equation (4) is relevant to increasing nonlinear diffusivity, as well as to decreasing diffusivity. After the substitution $u^* = C - S$ ($C \geq 1$, constant), we have the porous medium equation, with diffusivity that increases with dimensionless relative saturation $S$ (e.g., [10,11]),

$$S_t = \nabla \cdot \left[ \frac{1}{C - S} \nabla S \right]. \tag{5}$$

Since the remainder of the analysis is concerned with the form of exact solutions, the asterisk will be omitted when convenient. Hence, without loss of generality, one may assume $D_0 = 1$ in Equation (1). For appropriate non-dimensionalization of porous media flow equations, see, e.g., [12]. In one space dimension, the fact that the nonlinear diffusion equation with reciprocal square diffusivity $D = 1/(C - S)^2$ ($C \geq 1$, constant) is fully integrable has been used extensively in soil-water flow (e.g., [13,14]). Munier et al. [15] showed that in one space dimension, the reciprocal linear diffusivity in (5) is also special because it is invariant under the same reciprocal transformation [16] that linearises the fully-integrable model.

It is well known (e.g., Table 12 of [5]) that (1) has a free solution of the Cauchy–Riemann (CR) equations in its point symmetry group. Up to equivalence transformations, the Lie algebra of infinitesimal point symmetry transformations of the general nonlinear diffusion equation in $1 + 2$ dimensions is a general linear combination $\Gamma$ of the generators of translations by $(\alpha_0, \alpha_1, \alpha_2)$ in $(t, x, y)$, planar rotations by angle $\alpha_3$ and Boltzmann scaling $(\hat{t}, \hat{x}, \hat{y}) = (e^{2\alpha_4} t, e^{\alpha_4} x, e^{\alpha_4} y)$.

$$\Gamma = \alpha_0 \frac{\partial}{\partial t} + \alpha_1 \frac{\partial}{\partial x} + \alpha_2 \frac{\partial}{\partial y} + \alpha_3 \left[ x \frac{\partial}{\partial y} - y \frac{\partial}{\partial x} \right] + \alpha_4 \left[ 2t \frac{\partial}{\partial t} + x \frac{\partial}{\partial x} + y \frac{\partial}{\partial y} \right]. \tag{6}$$

The operator $\Gamma$ is the generator of the symmetry transformation:

$$\begin{align}
\hat{t} = \quad e^{\Gamma} t \quad &= t + \alpha_0 + 2\alpha_4 t + \mathcal{O}(\alpha_0^2 + \alpha_4^2), \tag{7} \\
\hat{x} = \quad e^{\Gamma} x \quad &= x + \alpha_1 - \alpha_3 y + \alpha_4 x + \mathcal{O}(\alpha_1^2 + \alpha_3^2 + \alpha_4^2), \tag{8} \\
\hat{y} = \quad e^{\Gamma} y \quad &= y + \alpha_2 + \alpha_3 x + \alpha_4 y + \mathcal{O}(\alpha_2^2 + \alpha_3^2 + \alpha_4^2). \tag{9}
\end{align}$$

For the special case $D = u^{-1}$ assumed in (1), there are additional symmetries generated by:

$$\Gamma_\infty = A(x, y) \frac{\partial}{\partial x} + B(x, y) \frac{\partial}{\partial y} - 2u A_x \frac{\partial}{\partial u}, \tag{10}$$

where $(A, B)$ is an arbitrary solution to the Cauchy–Riemann equations:

$$A_x = B_y; \ A_y = -B_x, \tag{11}$$

which is equivalent to $A + iB$ being a complex analytic function $f(z)$ of $z = x + iy$. Solutions of the CR equations are connected by the group of conformal maps, which is the symmetry group of Laplace's equation $\nabla^2 A = 0$.

In principle, one may choose conjugate harmonic functions $A(x, y)$ and $B(x, y)$ and then find a full set of independent invariants of $\Gamma_\infty$ either by solving the invariant surface condition:

$$A u_x + B u_y = -2u A_x \tag{12}$$

or by solving Pfaff's characteristic equations:

$$\frac{dx}{A} = \frac{dy}{B} = \frac{du}{-2u A_x}. \tag{13}$$

From the Cauchy–Riemann equations, this is equivalent to the system:

$$\frac{-2A_x dx}{A} = \frac{-2B_y dy}{B} = \frac{du}{u},$$

(14)

which integrates to:

$$\log(|u|) = \log(A(x,y)^{-2}) + p(y,t) = \log(B(x,y)^{-2}) + q(x,t),$$

(15)

for some functions $p$ and $q$. By differentiating throughout, with respect to $t$, this implies:

$$p_t(y,t) = q_t(x,t) = h(t),$$

(16)

for some function $h$. Hence, for some functions $H, \omega$ and $v$, $p = H(t) + \omega(y)$ and $q = H(t) + v(x)$. From this, it is seen that $u$ can only be in separated form, $u = \psi(t)w(x,y)$. Substituting this separated form in the nonlinear diffusion equation,

$$\psi'(t) = \frac{1}{w}\nabla \cdot \left(\frac{1}{w}\nabla w\right) = k \text{ (const.)}$$

(17)

Hence, the time-dependent solutions that result from conformal invariance can only be of the form:

$$u = (t + t_0)w(x,y) \text{ or } u = (t_0 - t)w(x,y),$$

(18)

for some constant $t_0$.

## 3. Infinite-Dimensional Class of Exact Solutions

An infinite-dimensional symmetry group of a nonlinear equation $N[u] = 0$ that includes a free solution of a linear equation $L[u] = 0$ in the same number of variables has previously been used as strong evidence of the integrability of the former. Bluman and Kumei [17] provided an algorithm to determine the equivalence transformation between an integrable nonlinear equation and a linear equation. Now, PDE (1) in three independent variables is conditionally integrable since its Lie symmetry group has an arbitrary solution of Laplace's equation in only two independent variables. From (18), the system consisting of (1) along with the side condition:

$$u_t = \frac{u}{t + t_0}$$

(19)

is in fact integrable. It would be helpful to find an explicit mapping from an arbitrary solution of Laplace's equation to a special class of t-dependent solutions of (1). A similar construction in [18] mapped a free solution of the Helmholtz equation in $n$ dimensions to a class of t-dependent solutions of nonlinear reaction-diffusion equations in $n + 1$ independent variables in any number $n$ of space dimensions. In that case, the solutions had exponential time dependence rather than linear time dependence.

First recall Liouville's mapping from a complex meromorphic function $f(z)$ to a solution:

$$V = \ln\left(\frac{8f'(z)\bar{f}'(\bar{z})}{[f\bar{f}(\bar{z}) + 1]^2}\right),$$

(20)

of the fully-integrable nonlinear PDE known as Liouville's equation [19],

$$\nabla^2 V + e^V = 0.$$

(21)

The complete general solution to Liouville's equation was given by Crowdy [20], in terms of two independent complex analytic functions. It is natural then to ask if the nonlinear diffusion

Eequation (1) is directly related to Liouville's integrable equation, after some extra constraint is added. Now, let $w = e^V$, giving:

$$- w = \nabla \cdot [w^{-1} \nabla w]. \tag{22}$$

Let:

$$u = (t_0 - t) w(x, y), \tag{23}$$

where $t_0$ is constant. Then, $-w = u_t$, so this gives an infinite-dimensional class of time-dependent solutions of (1), including a free complex analytic function $f(z)$. Explicitly,

$$\begin{aligned} u &= 8(t_0 - t) \frac{|f'(z)|^2}{(1 + |f(z)|^2)^2} \tag{24} \\ &= 8(t_0 - t) \frac{A_x^2 + B_x^2}{(1 + A^2 + B^2)^2}. \tag{25} \end{aligned}$$

In polar coordinates $z = re^{i\phi}$,

$$\begin{aligned} f(z) &= U(r, \phi) + iW(r, \phi) \\ U_r &= \frac{1}{r} W_\phi; \ W_r = -\frac{1}{r} U_\phi, \\ u &= 8(t_0 - t) \frac{U_r^2 + W_r^2}{(1 + U^2 + W^2)^2}. \tag{26} \end{aligned}$$

## 4. Properties of Solutions

With $t_0 > 0$, (26) allows for a broad class of positive initial conditions that vanish as $r \to \infty$. Although the solutions $u(\mathbf{r}, t)$ depend on both time and space variables, the flux density depends only on location and is constant in time. The solutions are consistent with prescribed-flux boundary conditions on any boundary curve. The flux divergence and therefore $u_t$ is constant at each location. Uniform extinction ($u = 0$) occurs at time $t_0$ due to the constant rate of net outflow through any complete boundary. In some examples, the boundary consists of the outer far-field $r \to \infty$, along with a small inner contour around a line source or line sink.

### 4.1. Axisymmetric Solutions

Consider the power-law function $f(z) = \alpha z^p = \alpha r^p e^{ip\phi}$. When $p$ is not an integer, this is not single-valued on $\mathbb{C}$, since it is not $2\pi$-periodic in $\phi$. However, the corresponding solution (26) for $u$ is still valid. That is,

$$u = 8p^2 |\alpha|^2 (t_0 - t) \frac{r^{2p-2}}{(|\alpha|^2 r^{2p} + 1)^2}. \tag{27}$$

At large $r$, with $p > 0$, $u \sim r^{-2-2p}$, and with $p < 0$, $u \sim r^{-2+2p}$. In any case, $u \sim r^{-2-2|p|}$. The boundary condition at large distance $r = R$ may be approximated by $u = 0$. For the case of an electron gas diffusing from a line source such as an intense laser beam or heated filament, this boundary condition may be effected by an absorbing or neutralising material at an outer cylinder.

The radial component of flux density is:

$$j(r) = 2(1 - p) \frac{1}{r} + 4p|\alpha|^2 \frac{r^{2p-1}}{1 + |\alpha|^2 r^{2p}}. \tag{28}$$

For $p > 0$, there is a point source or sink at the origin, with total influx:

$$\lim_{r \to 0} 2\pi r \, j(r) = 4\pi (1 - p) \tag{29}$$

and a greater total outflux at the infinite horizon,

$$\lim_{r \to \infty} 2\pi r \, j(r) = 4\pi(p+1). \tag{30}$$

For $p > 1$, the negative value of influx at the origin represents a point sink. For $p < 0$, the total influx at the origin is $4\pi(p+1)$ and the total outflow at the horizon is $4\pi(1-p)$. For $p < -1$, this represents a sink at the origin. In all cases,

$$j(r) \sim 2(1 - |p|)r^{-1}, \quad r \to 0, \tag{31}$$

$$j(r) \sim 2(1 + |p|)r^{-1}, \quad r \to \infty. \tag{32}$$

Although the case $p = 0$ corresponds to a trivial constant solution $f(z) = \alpha$ of the Laplace equation, the corresponding limit solution of the nonlinear diffusion equation has non-trivial flux density $j(r) \sim 2r^{-1}$.

The parameter $p$ is directly related to the line source strength $Q$ by:

$$|p| = 1 - \frac{Q}{4\pi}. \tag{33}$$

By the equation of continuity, the rate of decrease of total mass in the whole region is given by:

$$dM/dt = -8\pi|p| \,. \tag{34}$$

Since the total mass is zero at time $t_0$, the total initial mass must be $8\pi|p|t_0$. Conversely, if the total initial mass is specified to be $M_0$, the time for absorption is $t_0 = M_0/8\pi|p|$. At time $t = 0$, a strong electron absorber is placed around the boundary $r = R$. As $R$ increases, the solution with boundary condition (32) closely approximates the solution with perfectly-absorbing boundary condition $u(R, t) = 0$.

The solution involves an extra parameter $|\alpha|$. $\alpha$ does not appear in the expressions for asymptotic flux at $r = 0, \infty$. It may be regarded as an additional shape parameter for the solution. For example, when $p$ is small and positive, the initial condition near $r = 0$ is $u \sim 8p^2|\alpha|^2 t_0 r^{2p-2}$, so the length scale for the decrease of density is close to $2\sqrt{2t_0}p|\alpha|$. For $p = 1$ and for $p = -1$, there is zero flux at the origin. All of the solutions have outflow of mass at the infinite horizon.

In the application of the electron gas, the mass unit is the electron mass, so $M_0$ is the number of electrons in a cylindrical region with large diameter $R$.

The non-negative solutions that follow from (25) decrease to $u = 0$ at time $t_0$. By a very similar construction, one arrives at solutions that increase in time from a zero initial condition. Solutions of the form $u = tw(\mathbf{r}) = te^V$ imply:

$$\nabla^2 V - e^V = 0, \tag{35}$$

for which Liouville's solution is:

$$V = \log\left(\frac{8|f'(z)|^2}{(1 - |f(z)|^2)^2}\right). \tag{36}$$

For example, from $f(z) = \alpha z^p$; $p \in (0, 1)$, the solution is:

$$u = 8p^2|\alpha^2|t(r^{1-p} - |\alpha|^2 r^{1+p})^{-2}. \tag{37}$$

This solution has a zero-flux boundary condition at the inner surface of a cylindrical container at:

$$r = r_0 = |\alpha|^{-1/p}\left(\frac{1-p}{1+p}\right)^{1/2p}, \tag{38}$$

and a line source at the origin, supplying mass at total rate $dM/dt = 4\pi(1-p)$. Conceivably, this line source could be along the beam of an ionising laser. The exact solution is plotted in Figure 1, where it validates a numerical solution from the MAPLE routine pdsolve [21]. Over the domain with $0.1 \leq r \leq 0.33$ where $u/t$ varies from 13–25, the numerical and exact solutions agreed to within $6 \times 10^{-5}$. However, the numerical PDE solver failed to recover the exact solution when the inner radius was less than 0.1.

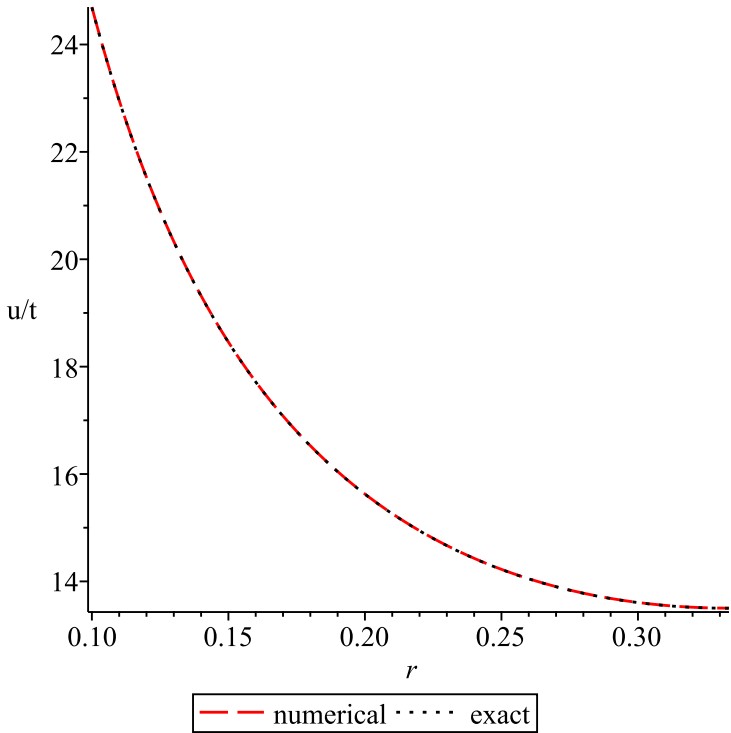

**Figure 1.** Radial flow from a point source to a container at $r_0 = 0.33$. $p = 0.5$. The analytic solution is compared to the numerical solution from the MAPLE routine pdsolve with flux boundary conditions prescribed at $r = 0.1$.

### 4.2. Image Source Solution

A solution with the no-flow boundary condition across the plane $x = 0$ may be simply constructed by the method of images. Take:

$$f(z) = \alpha([z-a]^p + [z+a]^p); \quad a \in \mathbb{R}^+. \tag{39}$$

Then, the construction (25) maintains reflection symmetry so there will be zero flow across the plane $x = 0$.

$$|f(z)|^2 = |\alpha|^2[(x^2 + y^2 - 2ax + a^2)^p + (x^2 + y^2 + 2ax + a^2)^p$$
$$+ 2\Re e(x^2 + y^2 + 2iay - a^2)^p]$$
$$|f'(z)|^2 = |\alpha|^2 p^2[(x^2 + y^2 - 2ax + a^2)^{p-1} + (x^2 + y^2 + 2ax + a^2)^{p-1}$$
$$+ 2\Re e(x^2 + y^2 + 2iay - a^2)^{p-1}]. \tag{40}$$

The construction (25) then gives a solution that is symmetric in both Cartesian axes, allowing no flow out of a quadrant. For example, with $p = 2$, the solution is a radial solution that is a one-parameter generalisation of (27),

$$u = 128|\alpha|^2(t_0 - t)\frac{r^2}{(1 + 4|\alpha|^2[a^4 + r^4])^2}. \tag{41}$$

*4.3. A Solution That Is Bounded on $\mathbb{R}^2 \times \mathbb{R}^+$*

The point source solution of Laplace's equation is the real part of:

$$f(z) = p \log z : \ p \in \mathbb{R}. \tag{42}$$

The real part is the potential of a radial source or sink, $A = p\Re e \log(r\, e^{i\phi}) = p \log r$. The conjugate harmonic function B is the imaginary part of the same function $f(z) = p \log z$, which is $p$ times the polar angle, $B = p\phi = p \arctan(y/x)$. This is also the real part of analytic function $f(z) = -ip\, \log(z)$.

$\nabla A$ gives a radial outward flow from a point source at the origin.

$\nabla B$ is a line vortex flow field (e.g., [22]). In the case of (42), the construction (26) results in flow lines that traverse all angles $\phi$, whereas the constructed solution $u$ is not a single-valued function that is $2\pi$-periodic in $\phi$. Following the idea of Popov [23], one way to rectify this is to replace $f(z)$ by:

$$F(z) = \sinh(f(z)) = \frac{1}{2r^p}[(r^{2p} - 1)\cos(p\phi) + i(r^{2p} + 1)\sin(p\phi)]. \tag{43}$$

With $F(z)$ replacing $f(z)$, the construction (26) then gives:

$$u = 32p^2 r^{2p-2}(t_0 - t) \frac{[r^{2p} - 1]^2 + [2r^p \cos(p\phi)]^2}{([r^{2p} + 1]^2 + [2r^p \sin(p\phi)]^2)^2}. \tag{44}$$

Furthermore, the solution is invariant under the replacement $p \to -p$, so without loss of generality, $p \in \mathbb{R}^+$.

The component of mass flux in the circumferential direction is $\frac{1}{ru}u_\phi$, which is zero at $\phi = 0, \pi/2p$. For practical purposes, the domain may be considered to be the wedge defined by $0 \le \phi \le \pi/2p$. Note that $u$ is $2\pi-$periodic in $\phi$ only if $p$ is an integer. However, when the domain has $\phi$ varying by less than $2\pi$, this restriction does not apply.

This solution is non-negative, $u \sim 32p^2(t_0 - t)r^{2p-2}$ as $r \to 0$ and diminishing rapidly, $u \sim 32p^2(t_0 - t)r^{-2p-2}$ at large $r$. For $p > 1$, the solution is bounded, attaining its maximum value $8(p^2 - 1)\left(\frac{p+1}{p-1}\right)^{1/p}$ at the wedge boundary $\phi = 0$ with $r = \left(\frac{p-1}{p+1}\right)^{1/2p}$. The minimum value zero is attained at the other wedge boundary $\phi = \pi/2p$ with $r = 1$, as well as at the origin $r = 0$. In the special case $p = 1$, the solution is bounded, with the maximum value $32(t_0 - t)$ attained at the origin. This solution represents a half-cylinder with a barrier at the flat boundary and rapid electron absorption taking place at a distant boundary $r = R$ ($R$ large).

Contours for the solution with $p = 1.5$ are shown in Figure 2.

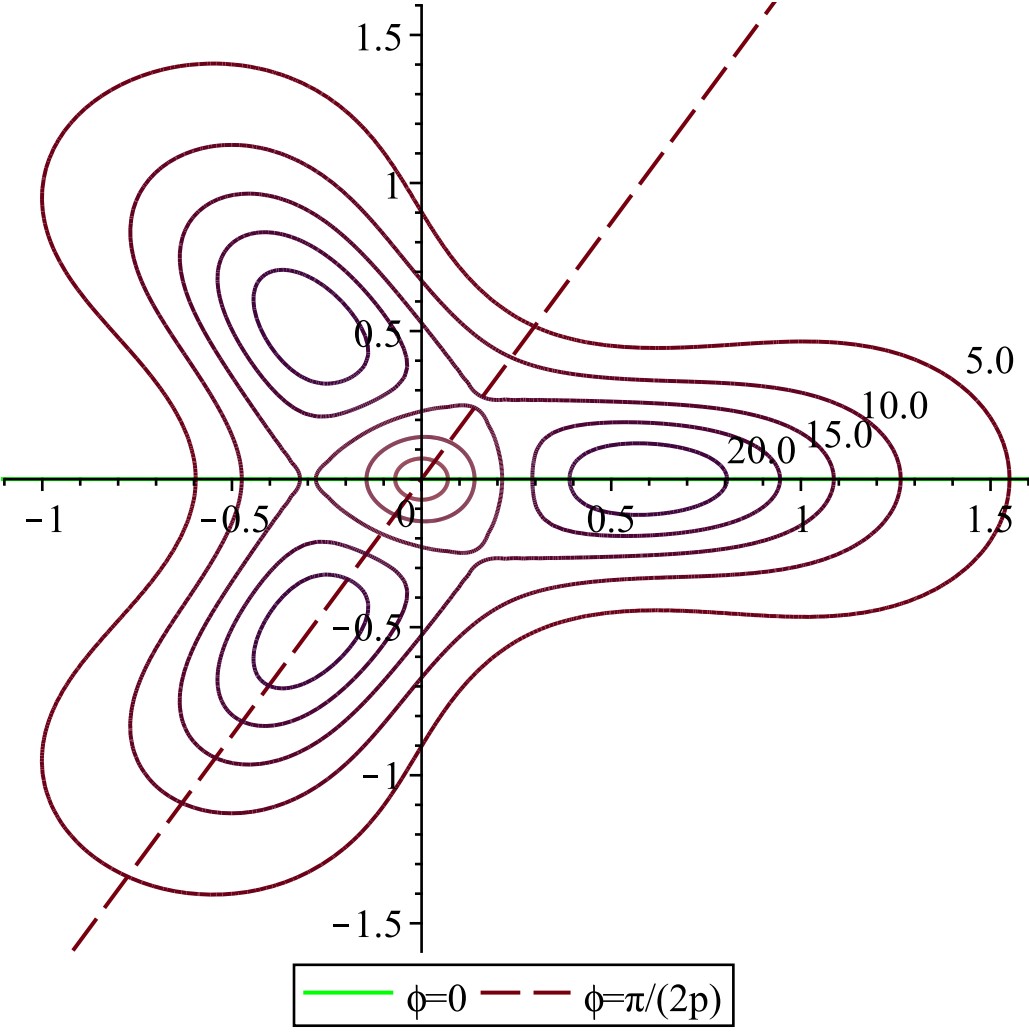

**Figure 2.** Contours of scaled density $u/(t_0 - t)$ from Equation (44) with $p = 1.5$. Contour levels vary from zero at $r = 0$ and $(r, \phi) = (1, \pi/3)$ to 29.24 at $(r, \phi) = (0.585, 0)$.

## 5. Conclusions

The 1 + 2-dimensional nonlinear diffusion equation with diffusivity varying inversely with a linear function of concentration is conditionally integrable in the sense that there is an explicit mapping from an arbitrary complex analytic function to an exact solution of the diffusion equation. These solutions have a simple linear time dependence and are self-similar. However, they obey time-independent flux boundary conditions that are meaningful in the experimental sense. The solutions provide insight into the nature of the transport process. For example, the solution (37) shows that if mass is supplied at rate $Q$ per unit time per unit length along an axis, with $0 < Q < 4\pi$ and with a concentric cylindrical barrier at any distance $r_0$, then there is a scale-invariant solution in which the concentration begins at zero initial condition, thereafter increasing linearly in time. From the constructed class of solutions, there is scope to impose a wide variety of geometric settings with prescribed flux boundary conditions on prisms of various cross-sections.

Liouville's solution of the steady exponential reaction-diffusion equation, which we have used here, uses a mapping from a free complex analytic function. In fact, the general solution of Liouville's equation is parametrised by two independent complex analytic functions [20]. This can lead to more general analytic solutions of the nonlinear diffusion equation, but their properties and applications have not yet been investigated.

**Author Contributions:** P.B. conceived of the idea of using Liouville's equation to solve the nonlinear diffusion equation and linked solutions to applications. J.M.G. completed most of the details of the calculations.

**Funding:** The first author is grateful for the support of the Australian Research Council through Project DP160101366.

**Acknowledgments:** We benefited from helpful discussions with Maureen Edwards and the late Nail Ibragimov.

**Conflicts of Interest:** The authors declare no conflict of interest.

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
