# Peer review of "Conditionally Integrable Nonlinear Diffusion with Diffusivity 1/u"

_symmetry, doi:10.3390/sym11060804_

Round 1
Reviewer 1 Report
The authors presented stationary solutions of nonlinear diffusion equation,
where a dynamical process is recombination only. Solution (8) immediately follows
from Eq. (1). However, it is proven that this is the only form for the class of stationary solutions.
It is a good paper and I recommend its publishing.
A marginal comment: nabla sign is missed in Eq. (3).
Reviewer 2 Report
The paper presents some mathematical analysis for the nonlinear diffusion equations with a varying diffusivity. The authors conclude that the nonlinear diffusion equation is conditionally integratabtle in the sense that there is an explicit mapping from an arbitrary analytic function to an exact solution of the diffusion equations.
It is an interesting research topic, but the authors should explicitly state what are the contributions of the paper?
(1) What is the background of the research?
(2) What is the domain on which the nonlinear PDE is defined?
(3) Could you draw some pictures of the solution under different conditions? This will help readers to understand the PDE properties.
(4) Could you possibly do some numerical experiments to verify your theory analysis?
(5) Could you introduce an application that benefits from your theory?
Minor comment:
(1) What is the $K_0$ on the first page?
Reviewer 3 Report
Referee report on “Conditionally Integrable Nonlinear Diffusion with Diffusivity 1/u” by P. Broadbridge 1,2*, J. M. Goard
The authors present an new mapping relevant to solutions of the nonlinear diffusion equations with a diffusivity that scales 1/u. The paper is well-written, conclusions are supported by the contents, and the abstract presents a concise summary. The application of this work is a physically motivated case, such that the paper has some appeal to the physics community, rather than just presenting a new mathematical solution. I would like the authors to consider, if they could improve the appeal of the paper by illustrating some results with figures, e.g. the plot the various solutions in u.
I found the discussion of the dimensions of equations 1 and 2 somewhat puzzling, as also the non-dimensionalization in section 2. Could the authors be more explicit in their steps? See also comment 4. I find this step to be very important if the mathematics is to be connected with a physical experiment.
Detailed comments appear below:
1) The time derivative in eqs. 1 and 2 is written in different forms. Could the authors use the same notation ? If they choose the index t, please state that this is the time derivative.
2) Could the authors provide a reference to the porous medium equation, possibly the first or just a review?
3) Apparently D0 is missing in equation 2, if one substitutes u=1-S as the authors write. Is this a bug, or could you comment what happened to D0? Later, the authors write that they can set it to one. But in my opinion the authors should not already do this here.
4) Could the authors comment on the dimensions of their quantities already when describing equations 1 and 2. Later in section 2 some comments appear, but they suggest that equations 1 and 2 are still with dimensions. Then the following inconsistency appears: u=1-S suggests that u and S are non-dimensional. Then (1) suggests that D0 is a diffusivity, [m^2/s] in SI units. As Ne has units [1/m^3], we see that D is in [m^5/s]. However, D=1/(C-S)^2 suggests it is non-dimensional, too. Furthermore, these observations do not seem to comply with the authors comments on the dimensions in section 2. Could the authors provide how to make physical sense of this somehow normalized equation?
5) In equation (3), the nabla* operator is defined. The dimensional nabla operator is missing on the RHS.
6) “… property of having an infinite dimensional Lie point symmetry group, even though it is not integrable.” Could you provide a reference?
7) Define gamma and the 4 alphas in the first equation after eq.4.
8) Also, I am not sure that I understand the rationale for numbering the equations. Why does equation 4 get an own equation identifier, and I am not sure why some equations are numbered and others are not numbered.
9) There are several grammatically wrong commas between subject and verb of a sentence. Remove the following commas: Page 3, sectrion3 in “…independent variables, is conditionally integrable…”; before and after “…, maps a free solution of the Helmholtz equation in n dimensions,…”; page 4 “… divergence and therefore du/dt, is constant…”; page 5: “…at the origin, represents…”; page 8 “…function of concentration, is conditionally integrable …”
Round 2
Reviewer 2 Report
All my comments are addressed. Thanks.